# Nanosized Calcium Phosphates as Novel Macronutrient Nano-Fertilizers

**DOI:** 10.3390/nano12152709

**Published:** 2022-08-06

**Authors:** Francisco J. Carmona, Antonietta Guagliardi, Norberto Masciocchi

**Affiliations:** 1Departamento de Química Inorgánica, Universidad de Granada, Av. Fuentenueva S/N, 18071 Granada, Spain; 2Institute of Crystallography and To.Sca.Lab., Consiglio Nazionale Delle Ricerche, Via Valleggio 11, 22100 Como, Italy; 3Dipartimento di Scienza e Alta Tecnologia e To.Sca.Lab., Università dell’Insubria, Via Valleggio 11, 22100 Como, Italy

**Keywords:** calcium phosphates, hydroxyapatite, ACP, nano-fertilizers, chemical doping

## Abstract

The need for qualitatively and quantitatively enhanced food production, necessary for feeding a progressively increasing World population, requires the adoption of new and sustainable agricultural protocols. Among them, limiting the waste of fertilizers in the environment has become a global target. Nanotechnology can offer the possibility of designing and preparing novel materials alternative to conventional fertilizers, which are more readily absorbed by plant roots and, therefore, enhance nutrient use efficiency. In this context, during the last decade, great attention has been paid to calcium phosphate nanoparticles (CaP), particularly nanocrystalline apatite and amorphous calcium phosphate, as potential macronutrient nano-fertilizers with superior nutrient-use efficiency to their conventional counterparts. Their inherent content in macronutrients, like phosphorus, and gradual solubility in water have been exploited for their use as slow P-nano-fertilizers. Likewise, their large (specific) surfaces, due to their nanometric size, have been functionalized with additional macronutrient-containing species, like urea or nitrate, to generate N-nano-fertilizers with more advantageous nitrogen-releasing profiles. In this regard, several studies report encouraging results on the superior nutrient use efficiency showed by CaP nano-fertilizers in several crops than their conventional counterparts. Based on this, the advances of this topic are reviewed here and critically discussed, with special emphasis on the preparation and characterization approaches employed to synthesize/functionalize the engineered nanoparticles, as well as on their fertilization properties in different crops and in different (soil, foliar, fertigation and hydroponic) conditions. In addition, the remaining challenges in progress toward the real application of CaP as nano-fertilizers, involving several fields (i.e., agronomic or material science sectors), are identified and discussed.

## 1. Introduction

Planet Earth has recently witnessed an enormous increase in the human population, predicted to reach almost 10 billion people by 2050 [1]. Feeding to a decent level all its inhabitants has become a global target, and, accordingly, the agriculture sector is asked to increase food production and optimize sustainable food production technologies [2,3]. These two goals can be simultaneously met if, beyond boosting the production performance of each cultivated land unit, the arable area is not lost by desertification, urban explosion, or industrial overbuilding. However, most agricultural soils are losing fertility and progressively degrade, due to climate changes and the application of exhaustive agricultural practices [4]. As a consequence, the rise in agricultural yields is necessarily linked to the intensification of crop production rather than to extending the cultivated areas [5]. In this regard, the massive use of agrochemicals (i.e., fertilizers and pesticides/herbicides) witnessed in the last decades causes eutrophication of aquifers, soil acidification, and release of greenhouse gases [3]. A numerical parameter addressing the effectiveness of a fertilizer impact is the Nutrient Use Efficiency (NUE), which is the ability of plants to suck nutrients from (typically) the soil at the root level and then transfer them to the shoot and other plant organs. Presently used N-based fertilizers have NUE values below 50%, thus implying that (more than) half of the total nitrogen supplied is lost in the environment [6]. In the case of phosphate, only a small fraction (5–30%) of the P supplied to crops is taken up by the plants [7]. Therefore, global sustainability requirements encourage the discovery/development of more efficient alternatives. In this respect, the development of new types of fertilizers might represent a suitable approach to intensifying agriculture production in a sustainable manner. Nanotechnological approaches optimizing the delivery of nutrients to crops, and limiting nutrient losses to the environment, can provide interesting novel solutions [8].

Nanotechnology offers the possibility of creating nanoparticles with sizes well below 100 nm, which, accordingly, show both a higher surface area and higher reactivity/solubility/dissolution rates than bulk materials. Suitably designed, they may thus feature higher efficiency and lower ecological risks at the expense of having higher production costs than their traditional counterparts [8]. Indeed, in recent years, the interest of crop fertilization stakeholders (industry, researchers, farmers, and governments) toward the development of nanomaterials as alternative fertilizers, known as nano-fertilizers, has considerably grown [9,10,11].

Nano-fertilizers are nanomaterials that can be composed of certain nutrients of interest, loaded with nutrients towards their sustained/controlled release, or both. Likewise, they can be designed to deliver macronutrients (nitrogen, phosphorus, potassium, and, to a lesser degree, calcium, sulfur, and magnesium) or micronutrients (i.e., iron, zinc, manganese, copper, among others) to crops. While plants need small amounts of micronutrients for optimal growth, crops require large quantities of macronutrients, and, consequently, industries produce fertilizers containing these nutrients in larger volumes. In addition, as indicated above, conventional macronutrient fertilizers, especially nitrogen- and phosphorous-based ones, are highly inefficient, and a great part of them are lost when supplied to crops, generating important economic, environmental, and health problems like eutrophication [10,12]. Surprisingly, despite the superior importance of macronutrient fertilizers, nanotechnology research in crop fertilization has mainly focused on the evaluation of nanomaterials as carriers of micronutrients, especially zinc, copper, manganese, and iron-based nanoparticles [11].

In this regard, calcium phosphate (CaP) nanoparticles, mainly nanocrystalline hydroxyapatite (nAp, ideal formula of Ca_5_(PO_4_)_3_OH) and, more recently, amorphous calcium phosphate (ACP, Ca_3_(PO_4_)_2_·nH_2_O), are a remarkable exception. During the last decade, these materials have raised a great interest in the development of macronutrient nano-fertilizers mainly due to their intrinsic composition in macronutrients (Ca, P) and their higher surface-to-volume ratio, enabling their functionalization with additional macronutrient-containing compound, like urea or nitrates. Therefore this paper reviews the advances in CaP as macronutrient nano-fertilizers. Firstly, the main characteristic of nanocrystalline apatite and amorphous calcium phosphate are briefly discussed. Later, the studies on the design and preparation of these materials as phosphorus- and nitrogen-nano-fertilizers are analyzed with special emphasis on the preparation and characterization of the materials as well as on their fertilization effects when supplied to crops.

## 2. Calcium Phosphate Nanoparticles: Nanocrystalline Hydroxyapatite and Amorphous Calcium Phosphates

Among the more than 60 different biologically-occurring minerals, CaP shows remarkable relevance since they are the main inorganic constituents of hard tissues in vertebrates and the exoskeletal structure of marine invertebrates [13,14]. Interestingly, biological CaP is always present as nanosized particles in calcified tissues, such as bones, teeth, or tendons of mammals. The nanometric size of CaP enables them to form a hierarchical structure with other biocomponents of the calcified tissues, like collagen and other non-collagenous proteins, to give stability and hardness to these organs while also acting as a reservoir of calcium and phosphate (and other) ions in the body [15,16].

Generally, CaP is present in living beings as non-stoichiometric nanocrystalline apatite, also known as biological apatite. Biological apatite is formed by tiny plate-like nanocrystals of highly structurally defective hydroxyapatite phase [13,17]. It contains relevant amounts of carbonate ions as substituents of PO_4_^3−^ and/or OH^−^ and, to a lower extent, potassium or sodium cations as substitutional defects of calcium sites. Remarkably, this chemical (non-stoichiometric) composition depends on calcium and phosphate/hydroxide ion deficiency and leads to increased solubility of the nanomaterial in aqueous media when compared with bulky stoichiometric apatite [18,19]. In addition, it is widely accepted that the existence of a more reactive, non-apatitic domain at the particle surface of biological apatite which shows an important role in several metabolic and biological functions like homeostasis or bone-regeneration [20]. In this regard, a recent exhaustive structural analysis assisted by Wide- and Small-angle X-ray (total)-scattering techniques (WAXTS and SAXS) have revealed that biogenic bone-apatite is composed of platy-like nanocomposites showing a core-crown arrangement, with a defective nanocrystalline apatitic core surrounded by a crown formed by an amorphous calcium phosphate phase (ACP) (Figure 1). Interestingly, it also proved the feasibility of preparing synthetic biomimetic apatite with similar chemical composition, size, morphology, and structure as biological apatite [17]. Therefore, the similarities of biomimetic apatite with biological apatite ensure the low toxicity and remarkable biocompatibility of the former. These properties, together with a large surface-to-volume ratio, favor the adsorption on the biomimetic apatite surface of relevant amounts of diverse compounds and have prompted researchers to exhaustively explore the applicability of these materials in the biomedicine field, including bone regeneration, implants-coating or as drug-delivery systems, among others [21].

Although to a lower extent than biological apatite, amorphous calcium phosphate nanoparticles (ACP) also form part of the mineral component of biological calcified tissues, like dental enamel or the exoskeletal structure of marine invertebrates [22]. ACP is considered a transient mineral precursor during hydroxyapatite biomineralization in both vertebrate and invertebrate organisms [22,23,24]. ACP is a highly disordered material that lacks long-range order, usually forming particles with spheroidal morphology [25,26]. It frequently shows a Ca/P molar ratio close to 1.5 (ideal formula Ca_3_(PO_4_)_2_·nH_2_O), although different Ca:P ratios can be found due to surface-adsorbed moieties or to the presence of occluded ions [27]. In addition, ACP is a metastable CaP phase in an aqueous solution, being spontaneously converted to other calcium phosphates, mainly non-stoichiometric apatite [28]. Nevertheless, the stability of ACP in water can be modulated by controlling the pH or by the usage of specific additives, like citrate and/or carbonate ions [29]. As in the case of biological apatite, ACP materials with similar features as in naturally-occurring amorphous calcium phosphate, namely size, morphology, and chemical composition, can be easily prepared following a biomimetic approach [30,31,32]. Thus, the application of these synthetic materials in the biomedical field has also been studied (i.e., the development of implant coating [33], cements and concrete, [34] and drug-delivery systems [35], among others).

## 3. Nanocrystalline Apatite as Macronutrient Fertilizer

Nanocrystalline apatite (nAp) shows specific features which make it an ideal candidate for the development of novel and more efficient macronutrient nano-fertilizers. These include (i) inherent content in macronutrients, namely calcium and phosphate; (ii) non-toxicity and biocompatibility; (iii) large surface-to-volume ratio due to its nanometric size, which enables it to adsorb relevant amount of compounds on the surface; (iv) aptitude to either cation or anion doping in its nanocrystalline structure; (v) pH-responsive solubility that opens the way to a controlled release of calcium and phosphate ions [36].

On the one hand, nanocrystalline apatite has been evaluated as potential phosphorous nano-fertilizers due to its large and inherent P-content. Conventional P-fertilizers, such as monoammonium or diammonium phosphate, are highly soluble salts easily available for plant uptake. However, a great part of them is fixed in the soil or lost by run-off after being supplied to crops, reaching the water bodies and generating severe environmental damages with unwanted societal and economic negative consequences [10,12]. To overcome these difficulties, phosphate sources formed by large particles (like naturally-occurring phosphate rocks of geological origin) have been proposed as an alternative to highly soluble P-fertilizers. However, the former is less effective in providing nutrients to the plants due to their low mobility, preventing phosphate from reaching the root zone and nurturing the crops in a timely fashion [37]. Alternatively, nano-sized apatite could be a trade-off between soil mobility and effectivity in P-nutrient release, thus minimizing the secondary contamination risks (Figure 2) [38].

In this regard, Liu and Lal evaluated the performance of nanocrystalline apatite as P-fertilizers. Specifically, they designed a nanocomposite based on nanocrystalline apatite coated with carboxymethyl cellulose (nAp particle diameter = 15.8 ± 7.4 nm). These new materials show long-lasting colloidal stability in aqueous media, an effect that could enhance their soil mobility [38]. In the same work, results are presented from a green-house experiment carried out to test the fertilization effect of synthetic nanocomposites in comparison with conventional triple superphosphate (TSP, Ca(H_2_PO_4_)_2_) on soybean (*Glicyne Max*). Interestingly, these results revealed that the nano-treatment promoted soybean growth to a larger extent than the conventional treatment (a 32.6% faster growth rate), leading to slightly enhanced crop yield and biomass production. This report suggested, for the first time, that nanocrystalline apatite could offer a superior phosphorus-use efficiency than highly soluble conventional P-fertilizers. In this context, Montalvo et al. also reported that nanocrystalline apatite (d = 20 nm) was a more efficient P-fertilizer than bulk-sized hydroxyapatite (d = 600 nm). However, it was not as efficient as commercial triple superphosphate (TSP) when supplied to wheat (*Triticum aestivum*) in Andisols (pH = 5.3–5.7) and Oxisols (pH = 5.9–6.4) soils, which are rich in iron and aluminum oxides and, therefore, strongly sorb phosphorus-containing compounds [39]. The authors firstly performed column transport experiments to evaluate the P-mobility of the three phosphorus-containing materials in both soils. In the case of Andisol, nanocrystalline apatite proved to show superior soil mobility than triple superphosphate compound and bulk apatite. Given the strong ability of this kind of soil to adsorb soluble phosphates, nAp could avoid soil fixation by moving as nanoparticles and reaching the plant roots where P-uptake mainly occurs. Instead, negligible nAp and apatite mobilities were observed in Oxisol. Regarding the fertilization studies, TSP proved to be the best fertilizer for P-uptake by wheat in both soils (64–88% of P in plants derived from TSP compound), followed by nanocrystalline apatite (40–61%) and bulk apatite (12–18%). These results suggest a superior capability of nanocrystalline apatite to provide phosphate to wheat plants than its bulk counterpart but a lower P-fertilization efficiency than conventional and highly soluble TSP fertilizer. In a subsequent study, the effectiveness of three different apatite nanoparticles, conventional TSP, and bulky phosphate rock (PR, mainly formed by bulky stoichiometric apatite) as P fertilizers in sunflower (*Helianthus annuus*) was analyzed. The plants were grown in two different P-deficient soils, namely an acid Ultisol (pH = 4.7) and an alkaline Vertisol (pH = 8.2) [40]. Firstly, nanocrystalline stoichiometric apatite (particle diameter = 25.7 nm) was synthesized by wet chemical precipitation (neutral nAp, zeta potential = −1.37 mV), and the surface of the nanoparticles subsequently functionalized with glycine or dibasic ammonium citrate to obtain positively surface charged-(nAp+, zeta potential = +22.1 mV) or negatively surface charged-nano-apatite (nAp−, −13.8 mV), respectively [41]. In acidic Ultisol soil, the treatments with TSP and the three different nanocrystalline apatites significantly improved plant growth in comparison with both PR and control treatments. This improvement was attributed by the authors to a marked increase in the shoot tissue P-concentration. Negatively surface charged nanopatite (nAp−) proved to be more effective in supplying P to sunflower plants growth in this kind of soil than TSP and other apatite-based treatments. The authors explained the superior behavior of nAp− based on two main causes. On the one hand, the slow and sustained release of phosphorus from a nanocrystalline apatite in the acidic soil decreased the amount of P binding to soil particles, which was lower than in the case of highly soluble TSP. Consequently, nAp− made it possible for a large amount of P to reach the root environment, enhancing the phosphorus supply to the plants. On the other hand, the negatively charged nAp− nanoparticles limited their fixation in the soil due to charge repulsion forces and, therefore, enhanced soil mobility in comparison with the other positively- and neutral-charged nanocrystalline materials. By contrast, TSP proved to be significantly the more efficient P-fertilizer in alkaline Vertisol soil due to the marked reduction of apatite solubility in basic media. Overall, these reports bring to light the need to design engineered nanocrystalline apatite as P-fertilizer with specific properties (particle size, surface charge, additives) matching the physical and chemical properties of the crop soil as well as of the targeted plants. In this way, the P-use efficiency of the novel engineered nano-fertilizers could be increased by reducing undesired nutrient losses. In this regard, it is important to highlight that all nanocrystalline apatite materials explored as P-fertilizers so far are nearly stoichiometric or with a limited degree of crystal defectiveness. However, it is possible to prepare defective nano-apatite (i.e., biomimetic apatite) by deliberately incorporating substitutional defects in its nanocrystalline structure and, hence, modifying its solubility. Consequently, the P-release profile from defective nano-apatite can be modulated toward the design of novel P-nano-fertilizers attending to their specific targeted application.

Nanocrystalline apatite has been also evaluated as carrier of N-containing compounds to obtain N-nano-fertilizers. Similarly to the employ of nanocrystalline apatite in the drug-delivery field [42], it has been proposed that the large surface area of the nanoparticles can be used to reversibly bind compound of interest (i.e., urea, nitrate ions) and release them gradually to favor nutrient uptake by plants.

For instance, in 2017, Kottegoda et al. prepared a urea-nAp nanohybrid by a two-step process: (i) wet-chemical precipitation of stoichiometric nanocrystalline apatite in an aqueous solution of urea and (ii) subsequent flash drying at 60 °C [43]. The resulting nanomaterials (rod-shaped 15–20 nm × 100 nm particles) proved to show a significant N-content (6:1 urea to nAp weight ratio). However, Powder X-Ray diffraction analysis revealed that the materials exhibited co-precipitated crystalline urea, leading to a conglomerate, i.e., a physical mixture of microcrystalline urea together with a nanostructured Ap powder carrying adsorbed urea molecules. In this regard, X-ray Photoelectron Spectroscopy (XPS) and Fourier-transform infrared spectroscopy (FTIR) analysis certified that the adsorbed urea molecules mainly interact with the Ca^2+^ sites on the nAp particles’ surface through their amino arms. Moreover, column transport experiments, using sand as substrate, showed that the urea-nAp interactions led to a slower release of the nutrient from the nanomaterial in comparison with pristine crystalline urea. Likewise, the crop-nutrition performance of the resulting nanomaterial was compared with granular, crystalline urea in a farm field during three different seasons, using rice as a crop model. The results showed that the treatment with the engineered-nano-fertilizer led to a significant superior nutrient use efficiency than the conventional one, resulting in a higher crop yield, namely 7.9 tons/hectare versus 7.3 tons/hectare, by adding half of the amount of the recommended N-nutrient value (50 kg N ha^−1^ and 100 kg N ha^−1^ for the nano- and conventional treatment, respectively). Authors associated the best performance of the nano-fertilizer with the slow-urea release shown by the nanomaterials, which avoided the decomposition/volatilization of urea in the soil and, therefore, favored nitrogen uptake by plants.

In a later report, a biomimetic chemical precipitation approach at 37 °C was employed to prepare novel CaP-based NPK-nano-fertilizers [31]. Specifically, citrate and carbonate ions, two important components of bone nano-apatite [44], were used as additives during the preparation of the nanoparticles, while calcium nitrate, potassium nitrate, and urea were employed as N- and K-sources. Transmission Electron Microscopy (TEM) together with Wide-Angle X-Ray Total Scattering (WAXTS) analysis certified that maturation times shorter than 24 h resulted in round-shaped fully amorphous calcium phosphate nanoparticles with diameters falling in the 10–25 nm range. By contrast, at longer maturation times, characteristic biomimetic nanocrystalline apatite with a platy shape was obtained (average T × W × L of 2.0 nm × 4.7 nm × 6.5 nm, T = thickness, W = width, L = length). Interestingly, elemental analysis and FTIR analysis revealed an abrupt drop in the N-payload concomitant with ACP-to-nAp transformation in the presence of both urea molecules and nitrate ions. These results indicate a superior loading capacity of ACP nanoparticles than of nanocrystalline apatite to host exogenous ions like nitrate and molecules like urea, prompting authors to select the doped-ACP nanomaterial to evaluate its performance as NPK-nano-fertilizer (*see next section*).

A similar finding was observed for biomimetic calcium phosphate nanoparticles prepared by wet chemical precipitation using only carbonate ions as an additive and Ca(NO_3_)_2_ and KNO_3_ as nitrate sources. In this case, while fully amorphous calcium-phosphate nanoparticles precipitated after 5 min of reaction with a 2.12% content of nitrate (weight basis), a nanocomposite made by ACP and nanocrystalline apatite was formed after 24 h of maturation time with a 0.44 % of nitrate [26]. A joint analysis of WAXTS and SAXS data indicated that the nanocomposite showed a nano-apatite-ACP core-crown arrangement, as it has also been suggested for biological apatite nanoparticles in bone [17]. Interestingly, WAXTS analysis revealed that the increase of the concentration of NO_3_^−^ precursors led to progressive *c*-axis expansion coupled with a concomitant *a*-axis contraction, which was attributed to the different amounts of nitrate incorporated as substitutional defects. This hypothesis was later corroborated by the nutrient release/dissolution profile (Ca^2+^, NO_3_^−^) shown by the nanoparticles when suspended in water (Figure 3a,b). Although a first burst NO_3_^−^ release was observed, associated with the anions weakly adsorbed to the amorphous crown, the nanocomposites showed a progressive nitrate release kinetic over (at least) 60 h. More interesting, the kinetic nitrate release profile was parallel to core-particle dissolution, confirming the incorporation of nitrate ions in the nanocrystalline structure. By contrast, the NO_3_^−^ kinetic profile from N-doped ACP was much faster than the amorphous nanoparticle dissolution, with a cumulative nitrate delivery of 94.2% after 10 h. In this regard, PXRD (Powder X-ray Diffraction) analysis certified that the amorphous nature of the nanoparticles was maintained during at least 60 h (Figure 3c), ruling out that ACP-to-nAp transformation elicits the nutrient release. Consequently, the nitrate-release process must be largely governed by the desorption of the dopant from the ACP particle surface, much more important than that generated by core particle degradation.

Overall, nanocrystalline apatite was proven to show good features to load nitrogen-containing compounds through two different processes: (i) surface adsorption of molecules and (ii) incorporation of ionic substituents in its nanocrystalline structure, in both cases leading to a slower nutrient release. However, while the former mechanism results in larger payloads of both ions and molecules, the second one conduces to poorer N-loading but to a more controlled nutrient release which is parallel to particle dissolution.

## 4. Amorphous Calcium Phosphate Nanoparticles as Macronutrient Nano-fertilizers

Much like nanocrystalline apatite, amorphous calcium phosphate (ACP) nanoparticles show intriguing characteristics which make them suitable candidates for developing novel macronutrients nano-fertilizers: nanometer size, large surface to volume ratio, elevated biocompatibility (due to their natural occurrence in some calcified tissues), intrinsic presence of macronutrients, and so on. In addition to that, ACP shows a higher solubility than nanocrystalline apatite (although still much lower than that of conventional fertilizers) and higher surface reactivity, enabling ACP to obtain larger nutrient payloads than nAp. Notwithstanding, the existing reports exploring the suitability of ACP as macronutrient nano-fertilizers are scarce and are mainly focused on the application of ACP as a carrier of N-compounds.

In 2020, Delgado-López et al. explored, for the first time, the feasibility of loading ACP nanoparticles with several macronutrients, namely nitrogen, phosphorus, and potassium (NPK), and employing the resulting engineered nanomaterials as slow-release nano-fertilizers in crops [31]. The amorphous nanoparticles were loaded with N- and K-compound by means of a *one-pot* approach, resulting in engineered nanomaterial (labelled as nano-U-NPK) composed of several macronutrients (values in parentheses are wt %): Ca (23.3), P (10.1), K (1.5) and N (2.6), with more than 80% of N in the form of urea molecules. The nutrient release studies in water confirmed an initial *burst* release for all macronutrients, attributed to the molecule/ions weakly adsorbed/occluded in the nanoparticles, followed by a more gradual release over more than 150 h. While only a small amount of the overall content in Ca, P, and K was released, the N-content was almost fully lost after 150 h, confirming that both urea molecules and nitrate ions were mainly desorbed from the particle surface. An additional column leaching assay in a simulated soil mimicking a low diffusive medium indicated that nano-U-NPK showed a urea release five times slower than conventional granular urea (Figure 4b), certifying the potential of the engineered-ACP nanoparticles as slow urea-release nano-fertilizers.

These encouraging results prompted the authors to evaluate the crop nutrition ability of the engineered nanoparticles. They first tested the fertilizing properties of nano-U-NPK by growing *Triticum durum* seeds in controlled environment chambers with this goal in mind [45]. Specifically, three different treatments were performed by supplying tap water or the conventional (NH_4_)_2_HPO_4_ fertilizer (DAP) and/or nano-U-NPK nanoparticles. After an initial DAP treatment of 36 kg of N ha^−1^, the untreated control group (1) received only water; the conventional fertilization group (2) received 150 kg of N ha^−1^ of granular DAP (as recommended for Andalusia and the Mediterranean area for cultivating wheat [46]), and the nano-U-NPK treated fertilization group (3) received only 15 kg of N ha^−1^ of a sprayed aqueous suspension of nano-U-NPK and 60 kg of N ha^−1^ of granular DAP (together with half of the N content than in group 2). Not unexpectedly, the untreated plants provided a ca. 40% lower kernel weight than groups 2 and 3, the latter two performing in a statistically similar manner. Shoot and ear numbers, kernel number, weight, and overall plant and ear weights behave similarly. More importantly, the amount of hard vitreous kernel and the protein content (well above 13% in the nanoU-NPK treatment) make these grains fall in the highest quality group for flour, semolina, and alimentary pasta products [47]. These results show the enhanced efficiency of nanoU-NPK within group 3, particularly once one considers that a 40% reduction of DAP treatment is found to be highly detrimental to kernel yield [48]. Additionally, experiments were conducted to identify the routes of nanoparticle uptake by staining the nanoparticles with Alizarin Red S, a specific dye for calcified animal and plant tissues [49]. Interestingly, nano U-NPK nanoparticles were up-taken by the plants through both roots and leaves. Although the absorption rate was significantly faster in the former than in the latter, this result is relevant because it confirms that nanoparticle dissolution (and consequently nutrient release) is not necessarily a limiting step to allowing nutrient uptake through the leaves. Noteworthy, these findings have also been observed for other kinds of nanoparticles [50,51].

In a separate report, the same engineered CaP-based nano-fertilizers were evaluated in products that intrinsically bear a much higher added value than wheat, specifically cv. Tempranillo grapes (*Vitis vinifera* L.). The tests on the fertilizing properties of doped nanosized amorphous calcium phosphate were performed by foliar spreading nano-U-NPK suspensions on commercial vineyards located at Instituto de Ciencias de la Vid y del Vino (ICVV-CSIC) in Logroño, Spain. In these field experiments, three different treatments of urea were realized by spraying over the grapevine leaves aqueous solutions of commercial urea at doses of 3 and 6 kg N ha^−1^ (here referred to as U3 and U6, respectively) or an aqueous suspension of nano-U-NPK (34.4 g L^−1^), consistent with a dose of 0.4 kg N ha^−1^ only. Additionally, a control experiment was performed by spraying plants with an aqueous solution of Tween© 80, also used as a wetting agent in all the treatments (1 mL L^−1^) [52]. The efficiency of the different fertilization protocols (U3, U6, and nano-U-NPK) was assessed by investigating the quality of the grapes, estimated by the YAN (yeast-assimilable nitrogen) and the chromatographic levels of amino acids, as these compounds are known to affect the grape quality and, therefore, determine the quality and health properties of the wine when the foliar application of the fertilizers is performed [53,54,55]. Interestingly, comparable, or even higher, values of YAN and amino N-content than U3 and U6 treatments are obtained using the nano-U-NPK material, despite its N content ca. 10× less than in the urea treatments. While urea loadings leading to N contents higher than 3 kg ha^−1^ are normally required for enhancing the aromatic quality of wine, most of the supplied fertilizer (as in the U3 and U6 treatments) is wasted, with obvious environmental and economic drawbacks, which the nano-U-NPK treatment could greatly mitigate.

Although the promising results demonstrated by nano-U-NPK ACP nanomaterials towards the efficient nutrition of different crops, such as wheat and vineyards, these CaP-based nano-fertilizers contain a moderate-to-poor N-payload (2.6 wt %). Likewise, the loading process of ACP with the N-compounds following the *one-pot* approach is highly inefficient, with significant losses of both urea and nitrate ions during the preparation process. To overcome these difficulties, an alternative post-synthetical modification strategy (PSM) to load urea molecules on ACP nanoparticles was later developed [56]. Interestingly, the PSM approach made it possible to nearly triplicate the nitrogen content of the nanomaterials (from 2.6 to 8.1 wt %) with no waste of the added urea by using 13.5 times less N-dopant than in the *one-pot* approach. PXRD and FTIR analysis revealed that the urea molecules were adsorbed onto the particle surface mainly by calcium-to-amine interactions, ruling out the co-precipitation of segregated phases of crystalline urea (Figure 5). The feasibility of producing the urea-doped nanoparticles by using tap water and inexpensive technical grade reagents was further proved, with the negligible detriment of the main properties while significantly reducing the manufacturing costs.

To evaluate the performance of the optimized U-doped nanomaterials (labelled as U-ACP), fertilization experiments on Pinot Gris variety grapevines were carried out [57]. Specifically, U-ACP nanoparticles, with a total amount of N being reduced by 20% compared to the conventional practice, where applied in fertigation or by foliar supply to crops, and their effect on the quality of the resulting grapes was compared with those obtained by using a conventional treatment with soluble NH_4_NO_3_ at the typical dose. As nitrogen availability has an influence on the formation of numerous compounds involved in the aroma matrix of wine [58,59,60], GVC profiles went carefully analyzed through multivariate statistical elaboration. The presence of a similar aromatic profile among the three different treatments agrees with what is described in Tempranillo grapes, where the N fertilization with proline, urea, and two commercial nitrogen fertilizers did not lead to an increase of primary grape aromas, such as terpenoids and norisoprenoids. In addition, the nano-fertigation treatments led to a larger YAN value than the conventional treatments, lower berry acidity (pH 7.4 vs. 7.1), and total soluble solid content (20.5 vs. 19.8 Brix), all these parameters being in line with the common consumer preferences. These observations suggest that the novel strategies based on U-ACP produce grapes with equivalent quality the those obtained by conventional treatments by significantly reducing the amount of N-supply and, therefore, showing a superior nutrient use efficiency.

On the other hand, recent strategies adopted in soilless agricultural farms include the development of vertically arranged plant growth, in which hydroponic or aeroponic conditions are installed. These farms are generally set in greenhouses or even indoors and are meant to save land in densely populated urban areas where horticultural consumption of fresh vegetables (lettuce, tomatoes, berries, etc.) is in high demand. Accordingly, a fertilization experiment with the optimized nano U-ACP was undertaken using hydroponically grown *Cucumis sativus* L. as a model. The fertilization protocol used in this case was as follows: after germination, plants were grown in a full nutrient solution for seven days and later kept under N starvation for additional seven days. Plants were then left under N starvation (as control) or treated with 1.0 and 0.5 mM in Nano-U-ACP (NP1.0 and NP0.5, respectively) and 1 mM urea (U1.0) [56].

The efficiency of the different fertilization treatments was assessed by investigating the cucumber plants’ fresh weight, as shoot and root biomasses. As expected, all NP1.0, NP0.5, and U1.0 treated samples increased the root biomass up to 100% in the NP0.5 case. At variance, shoot biomasses were not substantially affected, with a 10–20% increase observed in the N-supplied cases being (statistically) barely significant. A related analysis, i.e., the determination of the N concentration in the different portions of the plants, manifested a parallel behavior since more nitrogen is found in the roots of NP or Urea treated plants than in the control plants, and less differentiated values appear in the shoots. These observations suggest that biomasses and N-content are strictly related.

Deeper analysis also indicated that NP0.5 could indeed represent a more efficient (N saving) fertilization treatment as compared to 1.0 mM urea. Indeed, cucumber plants fed with NP0.5 displayed a nitrogen NUE of 0.69 ± 0.02, that was statistically much higher (*p* < 0.05) than the that determined for NP1.0 (0.48 ± 0.01) and for U1.0 (0.49 ± 0.01).

More recent research [61] aimed at understanding the molecular response of the urea uptake mechanism demonstrated that the slow release of urea from urea-loaded ACP nanoparticles could contribute to upregulating the urea uptake system for a longer period as compared to plants treated with bulk urea only. Indeed, such extended activation, studied by the expression analysis of the DUR3 gene (a transporter devoted to the transmembrane translocation of urea), was mirrored by the higher accumulation of N in NP-treated plants, even when the supplied urea concentration was halved (i.e., in the NP0.5 treatment) confirming; therefore, the higher performance previously described. These observations, therefore, beautifully link the physiological, phenomenological, and molecular levels in a coherent picture.

## 5. Conclusions

In conclusion, calcium phosphate nanoparticles, including nanocrystalline apatite (nAp) and amorphous calcium phosphate (ACP), have demonstrated intriguing features making them ideal candidates for developing novel nano-fertilizers with high efficiency. Both materials are intrinsically composed of macronutrients, namely calcium and phosphorus, which can be gradually delivered simultaneously with particle dissolution, limiting nutrient losses by soil fixation or run-off. Likewise, both materials can load other macronutrient-containing species like nitrate ions or urea. In this regard, while ACP has demonstrated to enable larger N-payloads due to its superior surface reactivity, nAp is able to incorporate nitrate into its nanocrystalline lattice as a vicariant ion, leading to slower nitrate-release profiles. In addition, several reports have also shown that the novel CaP-based fertilizers presented a superior nutrient-use efficiency than their highly soluble conventional counterparts in distinct cases (wheat, grapes, cucumbers) and through different ways of supply (i.e., foliar supply, fertigation, soilless cultivations).

## 6. Outlook

Despite the encouraging results that CaP has demonstrated as nano-fertilizers, the application of these nanomaterials in agriculture is still in its infancy, and additional advances are needed to properly evaluate the real potentiality of these nanomaterials in crop nutrition, with a specific emphasis on cost-reduction analysis toward their massive usage, an aspect which is still largely underexplored.

Concerning the material science aspects, the vast knowledge of the several calcium (ortho)phosphates acquired during the decades of research, especially in the biomaterial field, may help if applied to this emerging topic. Specifically, already existing or novel synthetic protocols enable an exquisite control of the relevant nanoparticle features (i.e., size, morphology, chemical composition, functionalization degree, and so on) and can sagaciously be employed in the preparation of suitably tailored calcium phosphate nanostructured materials. Likewise, the usage of several forefront structural and analytical techniques, like WAXTS or SAXS analysis, is essential to precisely characterize many physico-chemical aspects of these nanomaterials at multiple (atomic and nanometer) length scales. Concerning the agronomic field, it is essential to carry out further in-depth analysis to understand the physiological and molecular mechanism responses of plants to CaP nanoparticle supply. In this way, the finely tuned features of the nanoparticles could be firmly correlated with the elicited effects on crops, favoring the rational design and preparation of novel advanced CaP-based nano-fertilizers.

In summary, we hope that the promising results included in this short review will encourage researchers from multiple fields (chemistry, materials science, agronomy, engineers, economists, and investors) to combine their diverse expertise and work together toward the development of smart macronutrient CaP-based nano-fertilizers, fostering the progress and development of efficient and sustainable precision agriculture.

## Figures and Tables

**Figure 1 nanomaterials-12-02709-f001:**
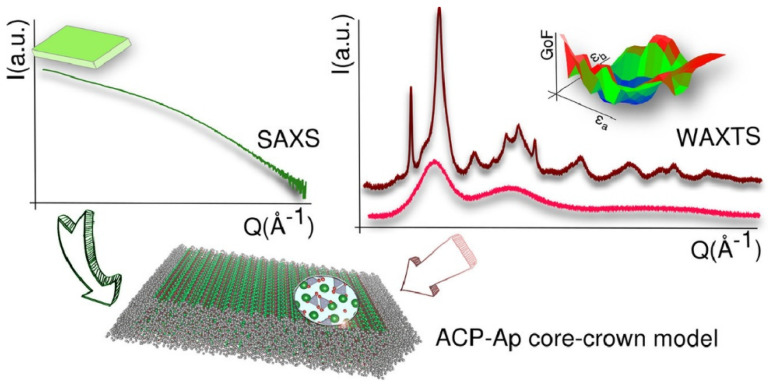
Schematic representation of the joint SAXS and WAXTS analysis used to elucidate the ACP-nAp core-crown arrangement of the nanocomposites platelets present in both biogenic and biomimetic apatite “Reprinted with permission from Ref. [17]. 2021, Elsevier”.

**Figure 2 nanomaterials-12-02709-f002:**
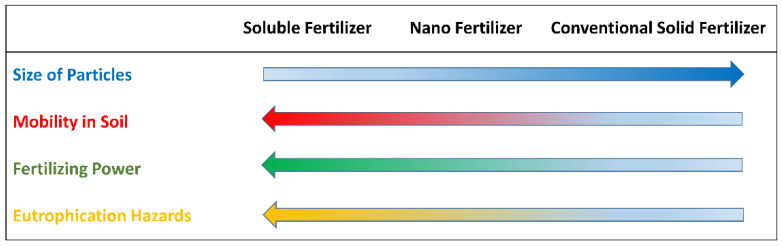
A schematic comparison of soluble P, nano-sized solid P and solid P fertilizers and their environmental properties. “Adapted with permission from Ref. [38]. 2014, Springer Nature”.

**Figure 3 nanomaterials-12-02709-f003:**
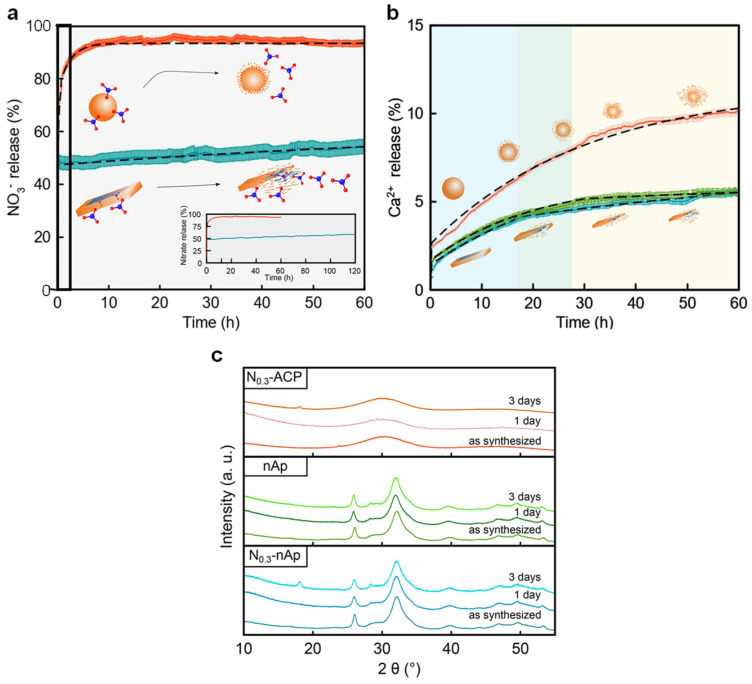
Cumulative release of nitrate (**a**) and calcium (**b**) from nitrate-doped nano-apatite (blue curve) and ACP (red curve) and non-doped nano-apatite (green curve); (**c**) Structural evolution of nitrate-doped nano-apatite, apatite and ACP nanoparticles after their suspension in water during days 1 and 3. “Adapted with permission from Ref. [26]. 2020, Springer Nature”.

**Figure 4 nanomaterials-12-02709-f004:**
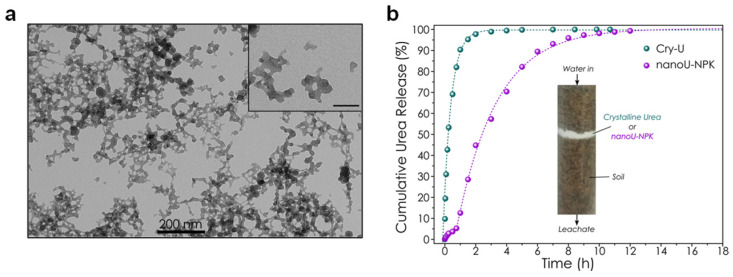
(**a**) TEM image of nanoU-NPK nanoparticles. (**b**) Cumulative urea release from granular urea (blue curve) and nanoU-NPK (magenta curve) embedded in a simulated solid medium, mimicking an inert soil (a low-diffusive medium) during water irrigation. The inset shows a graphical representation of the column used in the leaching experiments “Reprinted with permission from Ref. [31]. 2020, ACS”.

**Figure 5 nanomaterials-12-02709-f005:**
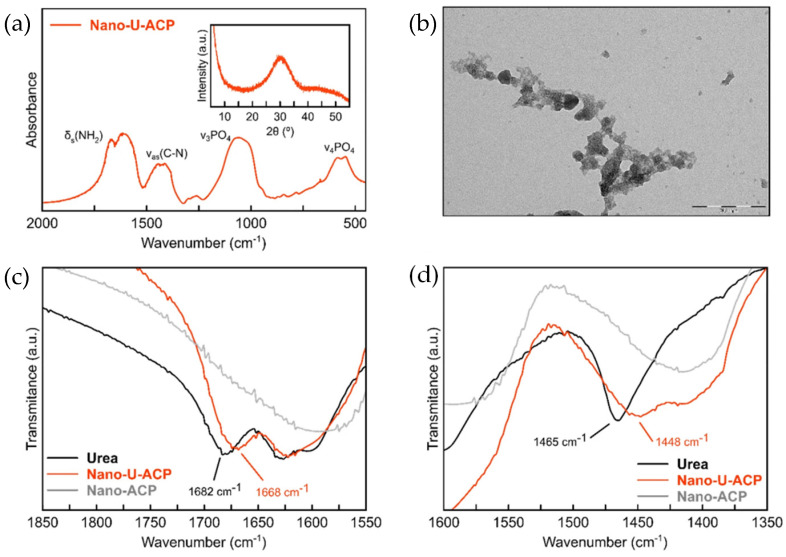
(**a**) FTIR and PXRD of Nano-UACP, (**b**) TEM-images of NanoU-ACP nanoparticles showing spheroidal morphology, (**c**,**d**) FTIR spectra of Urea (black curve), Nano-U-ACP (red curve), and non-functionalized ACP (Nano-ACP, grey curve) in the regions of wavenumber from 1850–1550 cm^−1^ and 1600–1350 cm^−1^, respectively. The shift observed in the vibration mode assigned to the urea for the material Nano-U-ACP and pristine urea indicating the main Ca-NH2 interactions are highlighted. “Reprinted with permission from Ref. [56]. 2021, Springer Nature”.

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
