# Peer review of "Nanosized Calcium Phosphates as Novel Macronutrient Nano-Fertilizers"

_nanomaterials, 2022, doi:10.3390/nano12152709_

Round 1

Reviewer 1 Report

The progresses of nanosized calcium phosphates as novel macronutrient nanofertilizers were reviewed. The topic is interesting, and the manuscript is informative and attractive. After moderate revision, I think it can be accepted.

1.    The abstract should be rewritten. The abstract needs to briefly summarize the content of the full text. However, in this manuscript has 4/5 sentences for background explanation, and only 1/5 sentences to describe the content of the text.

2.    Even in the reviewing article, it is not recommended to quote a lot of other people's charts. The five figures in this manuscript are all reproduced from references. It is suggested the authors to redesign and redraw 1-2 representative figures and delete unnecessary figures.

3.    The introduction of some case studies in a reviewing manuscript should be highly summarized, and it is not suitable to introduce some studies in details from the research significance, methods, to results and discussion. Therefore, the citation of literature 26, 44, 59, 60 must be simplified.

Author Response

Reviewer 1

The progresses of nanosized calcium phosphates as novel macronutrient nanofertilizers were reviewed. The topic is interesting, and the manuscript is informative and attractive. After moderate revision, I think it can be accepted.

We thank the reviewer for his/her positive comments.

  1. The abstract should be rewritten. The abstract needs to briefly summarize the content of the full text. However, in this manuscript has 4/5 sentences for background explanation, and only 1/5 sentences to describe the content of the text.

The abstract has been largely modified and a summary of the content of the paper, jointly with some concluding remarks, complements the original version in a more organic manner. This also answers a concern of Reviewer # 2.

  1. Even in the reviewing article, it is not recommended to quote a lot of other people's charts. The five figures in this manuscript are all reproduced from references. It is suggested the authors to redesign and redraw 1-2 representative figures and delete unnecessary figures.

Out of five figures, four are from our own research, and we got regular permission for publishing them in this review. Only Figure 2 (with permission) is taken from other people’s work and, in the new version, has been replaced by a fully new chart, redrawn by us as requested by the reviewer.

  1. The introduction of some case studies in a reviewing manuscript should be highly summarized, and it is not suitable to introduce some studies in details from the research significance, methods, to results and discussion. Therefore, the citation of literature 26, 44, 59, 60 must be simplified.

The reviewer is absolutely right and, according to his third comment, The pertinent paragraphs have been shortened and unnecessary details and technicalities have been eliminated. The revised text is greatly simplified.

Reviewer 2 Report

This is a very interesting study, it should be of interest to readers, but there is some information to be added before publication.

1. The abstract needs to be supplemented with specific conclusions, not just what studies have been done.

2. It is recommended to write the conclusion and outlook in two parts.

Author Response

Reviewer 2

This is a very interesting study, it should be of interest to readers, but there is some information to be added before publication.

We thank the reviewer for his/her positive comments.

  1. The abstract needs to be supplemented with specific conclusions, not just what studies have been done.

The abstract has been largely modified and a summary of the content of the paper, jointly with some concluding remarks, complements the original version in a more organic manner. This also answers a concern of Reviewer # 1.

  1. It is recommended to write the conclusion and outlook in two parts.

Conclusions and Outlook appear now in two separate sections.

Round 2

Reviewer 1 Report

Although the four figures are from your own research, I don't think it is necessary to reproduce them in this review.